# Characterization of Arsenite-Oxidizing Bacteria Isolated from Arsenic-Rich Sediments, Atacama Desert, Chile

**DOI:** 10.3390/microorganisms9030483

**Published:** 2021-02-25

**Authors:** Constanza Herrera, Ruben Moraga, Brian Bustamante, Claudia Vilo, Paulina Aguayo, Cristian Valenzuela, Carlos T. Smith, Jorge Yáñez, Victor Guzmán-Fierro, Marlene Roeckel, Víctor L. Campos

**Affiliations:** 1Laboratory of Environmental Microbiology, Department of Microbiology, Faculty of Biological Sciences, Universidad de Concepcion, Concepcion 4070386, Chile; constazherrera@udec.cl (C.H.); bbustamante2017@udec.cl (B.B.); claudiavilo@gmail.com (C.V.); paulinaaguayo@udec.cl (P.A.); crvalenz@gmail.com (C.V.); csmith@udec.cl (C.T.S.); 2Microbiology Laboratory, Faculty of Renewable Natural Resources, Arturo Prat University, Iquique 1100000, Chile; 3Faculty of Environmental Sciences, EULA-Chile, Universidad de Concepcion, Concepcion 4070386, Chile; 4Institute of Natural Resources, Faculty of Veterinary Medicine and Agronomy, Universidad de Las Américas, Sede Concepcion, Campus El Boldal, Av. Alessandri N°1160, Concepcion 4090940, Chile; 5Faculty of Chemical Sciences, Department of Analytical and Inorganic Chemistry, University of Concepción, Concepción 4070386, Chile; jyanez@udec.cl; 6Department of Chemical Engineering, Faculty of Engineering, University of Concepción, Concepcion 4070386, Chile; victorguzmanfierro1@gmail.com (V.G.-F.); mroeckel@udec.cl (M.R.)

**Keywords:** arsenic, arsenite-oxidizing, biofilm, detoxification

## Abstract

Arsenic (As), a semimetal toxic for humans, is commonly associated with serious health problems. The most common form of massive and chronic exposure to As is through consumption of contaminated drinking water. This study aimed to isolate an As resistant bacterial strain to characterize its ability to oxidize As (III) when immobilized in an activated carbon batch bioreactor and to evaluate its potential to be used in biological treatments to remediate As contaminated waters. The diversity of bacterial communities from sediments of the As-rich Camarones River, Atacama Desert, Chile, was evaluated by Illumina sequencing. Dominant taxonomic groups (>1%) isolated were affiliated with *Proteobacteria* and *Firmicutes*. A high As-resistant bacterium was selected (*Pseudomonas migulae* VC-19 strain) and the presence of *aio* gene in it was investigated. Arsenite detoxification activity by this bacterial strain was determined by HPLC/HG/AAS. Particularly when immobilized on activated carbon, *P. migulae* VC-19 showed high rates of As(III) conversion (100% oxidized after 36 h of incubation). To the best of our knowledge, this is the first report of a *P. migulae* arsenite oxidizing strain that is promising for biotechnological application in the treatment of arsenic contaminated waters.

## 1. Introduction

Arsenic (As) is a metalloid present in the atmosphere, soil and water as the result of various natural processes, such as land erosion and volcanic emissions, as well as the consequence of anthropogenic activities [1]. This element is present in many natural environments, typically in the form of inorganic As combined with other elements, such as oxygenated and sulfate-rich mineral waters [2,3,4]. In neutral pH water, two major forms of inorganic As can be found: the reduced form as trivalent arsenite and the oxidized form as pentavalent arsenate. Both the reduced and oxidized forms of As are toxic, but arsenite is on the average 100 times more toxic and it is also more mobile than arsenate [5,6,7].

The toxicity of arsenite is due to its strong binding affinity for sulfhydryl groups in proteins, thus affecting the redox status of the cysteine residues present on active sites of many enzymes and impairing their activities [5,8]. Arsenite also reacts with dithiol groups of glutathione, glutaredoxin and thioredoxin, interfering with intracellular redox homeostasis, DNA synthesis and repair and protein folding [8]. On the other hand, since arsenate is a natural analogue of the phosphate molecule, it can compete with phosphate in various biological processes [5,8,9,10]. In addition, arsenate uncouples oxidative phosphorylation, disrupts ATP synthesis and impairs various ATP-dependent cellular processes, such as transport, glycolysis, the pentose phosphate pathway and signal transduction pathways [11]. The chronic exposure to As (for example in drinking water) may cause severe health problems. The International Agency for Research on Cancer, the United States Department of Health and Human Services and the United States Environmental Protection Agency have classified inorganic arsenic as carcinogenic for humans, increasing the risk of skin, liver, bladder and lung cancers [12,13,14,15].

As-contaminated drinking water has been identified in different parts of the world, including Bangladesh, China, India, Pakistan and the U.S.A., affecting millions of people due toxicity [2,16,17]. In Chile, the problem of chronic arsenicism affects around 50,000 persons, mainly in rural populations of the Atacama Desert [18,19]. The affected populations drink water from small waterfalls and rivers with arsenic contents greater than 1000 ug/L [20,21]. This situation greatly surpasses both the World Health Organization and the U.S. Environmental Protection Agency recommendations for As concentrations (up to 10 ug/L) [22].

As-removal by conventional technologies (electro/coagulation, filtration, lime softening, activated alumina adsorption, ion exchange, reverse osmosis, electrodialysis reversal and nanofiltration) is a process able to remove nearly 80–95% of the As. These methodologies generally make use of a two-step method involving the oxidation of As(III) followed by the adsorption of the As(V) produced [23,24,25,26]. However, the strong chemical oxidants used in these processes are high-priced and may generate secondary pollutants [3,16].

Several metabolic pathways are involved in bacterial arsenate reduction (*ars*ABC and *arr*AB) and arsenite oxidation (*aio* and *arx*), both under aerobic and anaerobic conditions [6,8,12]. In particular, arsenite oxidase, enzyme catalyzing As(III) oxidation has been characterized in both autotrophic and heterotrophic bacteria, and contains two heterologous subunits: a large catalytic subunit (AioA) that contains the molybdenum cofactor together with a 3Fe–4S cluster, and a small subunit (AioB) that contains a Rieske 2Fe–2S cluster [4,8]. Microorganisms inducing oxidation of arsenite in aqueous phase into the less mobile and less toxic arsenate has been known for many years and members of no less than nine bacterial genera, including alpha, beta and gamma *Proteobacteria*; *Deinocci* (*Thermus*) and *Crenarchaeota*, have being involved in this process [27,28,29,30,31]. Thus, biological oxidation of arsenite has the potentiality to bioremediate As-contaminated waters, making it a promising alternative to the traditional chemical oxidants. Hence, bacteria capable to oxidize arsenite must be considered as at least part of the solution. Therefore, the aim of this study was to characterize natural community arsenic resistant of Camarones river and isolate a bacterial strain resistant to As and to evaluate its capability to oxidize arsenite in planktonic and sessile (biofilm) conditions. All this data is required to consider the possible use of an As-resistant selected bacterial strain to implement a biological treatment system to bioremediate arsenite contaminated water.

## 2. Materials and Methods

### 2.1. Sampling

Sediments samples were collected from the Camarones river (18°59′46.8″ S 69°46′33.0″ W), in triplicate, from the surface up to a depth of 10 cm using 15 cm long and 3 cm diameter sterile cores. After the collection, the sample was maintained at 4 °C while transported to the laboratory. The superficial layer, 2 cm, of each sample was mechanically homogenized within a laminar flow hood (ZHJH-C 1109C, Zhicheng, Korea) and stored at 4 °C in RNAlater (Invitrogen-Thermo Fisher Scientific, CA, USA) for further processing.

### 2.2. Analysis of the Bacterial Community Composition

Total DNA from 1 g of the upper 2 cm layer of the sediment sample (mix of the three replicas) was extracted using the UltraClean DNA extraction kit (MO BIO Laboratories Inc., Carlsbad, CA, USA) following the protocol provided by the manufacturer. DNA extracts were then purified. The quality and concentration of DNA was checked by UV/Vis spectroscopy (NanoDrop ND-1000, Peq-lab, Erlangen, Germany). The DNA was sequenced at the Greehey Children’s Cancer Research Institute Next Generation Sequencing Facility, UT Health Science Center, University of Texas System, San Antonio, TX, USA. Sequencing was achieved using the Illumina MiSeq technology with pair end reads, targeting the V1-V2 region of 16S rRNA with primers 27F/338R (forward primer: 5′-AGA GTT TGA TCM TGG CTC AG-3′, reverse primer: 5′-GCT GCC TCC CGT AGG AGT-3′).

The Mothur software v.1.33.3 (www.mothur.org (accessed on 10 December 2020)) was used for pre-processing the sequenced data, including quality check, removal of duplicates, alignments to SILVA database (http://www.arb-silva.de (accessed on 10 December 2020)), denoising and chimera removal. Then, Mothur was used for distances calculation and clustering of the sequences into OTUs with a distance of 0.03. For taxonomic assignment, the RDP classifier version 2.7 and SILVA database with a threshold of 0.8 were used. For statistical analysis, we used the R statistical software version 3.1.0 [32].

### 2.3. Isolation of Arsenic Resistant Bacterial Strains

One g of the top 2 cm layer of the sediment was inoculated into 50 mL chemically defined medium (CDM) [33] (pH 8.0) containing 100 mg L^−1^ Na_3_AsO_3_ (Sigma-Aldrich, San Luis, MO, USA) and incubated at 30 °C for 48 h, on a shaker adjusted at 130 rpm, in darkness, according to Campos et al. [34]. Bacteria were then isolated in and CDM agar plates supplemented with arsenite (0.5 mM). The plates were incubated at 30 °C, for 48 h, in darkness.

### 2.4. Tolerance Levels to Arsenic (TLA)

The Tolerance levels to arsenic (TLA) for the isolates was determined by the agar dilution technique, on Luria Bertani (LB) agar (Becton Dickinson and Company, Maryland, MO, USA) plates supplemented with different concentrations of sodium arsenite (0.5–100 mM) (Sigma-Aldrich, San Luis, MO, USA) or sodium arsenate (0.5–1000 mM) (Sigma-Aldrich, San Luis, USA). Each plate was inoculated with cell suspensions from fresh pre-cultures at a final density of approximately 10^7^ colony forming units (CFU) mL^−1^ and incubated for 24 h at 25 °C. To be used as control, LB agar plates were seeded with bacteria but without the metalloid. The criterion to consider bacterial strains as resistant or sensitive to As(III) or (As(V) was the one described by Rokbani et al. [35]. The bacterial strain capable to tolerate the highest arsenite and arsenate concentrations was selected to perform all the following experiments.

### 2.5. Identification of the Selected Strain and Aio Gene Detection Based on Molecular Characterization

The As resistant bacterial strain selected was identified after polymerase chain reaction (PCR) amplification of its 16S rRNA gen. The DNA was extracted using the InstaGen Matrix kit (BIO-RAD, Hercules, CA, USA) following the directions of the manufacturer. PCR was done using 16S rDNA bacteria universal primers GM3 (AGAGTTTGATCMTGGC) and GM4 (TACCTTGTTACGACTT). The PCR procedure followed the description of Urdiain et al. [36]. To detect the arsenite oxidase, a PCR analysis was performed using degenerated primer which target the large subunit of arsenite oxidase 69F (5′-TGYA TYGT NGGN TGYG GNTA YMA-3′) and 1374R (5′-TANC CYTC YTGR TGNC CNCC-3′), which target the large subunit of arsenite oxidase, according to Valenzuela et al. [21]. Sequencing was conducted by means of the Sanger’s method using an ABI PRISM 3500 xL Genetic Analizer (Applied Biosystems, Foster City, USA). The sequences were analysed by means of Basic Local Alignment Search Tools (BLAST) (http://www.ncbi.nlm.nih.gov/BLAST (accessed on 10 December 2020)) to obtain sequences with the greatest significant alignment.

### 2.6. Effect of As(III) and As(V) on the Growth of the Selected Bacterial Strain

The selected bacterial strain (~10^4^ CFU mL^−1^) was inoculated into 50 mL CDM medium supplemented with 0.5 mM As(III) or As(V) and then incubated under aerobic conditions at 30 °C under agitation (130 rpm) for 48 h. A control was implemented without the addition of As and incubated under the same conditions described above. The experiments were performed in triplicate and growth was determined recording the number of CFU at regular time intervals. The curves obtained were analysed, making use of the mathematical modelling of Gompertz [37]. Graphs and models were made using GraphPad Prism version 5.0 (GraphPad software, San Diego, CA, USA).

### 2.7. Attachment of the Selected Strain to Inert Support (Biofilm Formation)

The biofilm was formed placing sterile activated carbon and CDM (5:100 *w*/*v*) in 250 mL flasks. Then, an inoculum (10^5^ CFU mL^−1^) of the selected strain was added and cultured at 30 °C, with agitation (130 rpm), for 72 h under aerobic conditions, in the presence or absence of 0.5 mM As(III). The experiments were performed in triplicate and biofilm formation was evaluated by total cell counts using fluorescence microscopy and by scanning electron microscopy (SEM).

### 2.8. Scanning Electron Microscopy (SEM)

Samples of activated carbon were fixed using 2.5% glutaraldehyde (Sigma-Aldrich, San Luis, USA). Samples were dehydrated with ethanol and critical point dried using CO_2_. Then, they were coated with gold using an Edwards S150B sputter coater (Edwards High Vacuum International, West Sussex, UK) [38]. Samples were analysed using a JEOL JSM 6380LV SEM (JEOL, Tokyo, Japan).

### 2.9. Fluorescence Microscopy

One g of activated carbon samples were sonicated in 10 mL phosphate-buffered saline (PBS) (pH 7.0) in an ultrasonic bath (Elma/S30, Singen, Germany) adjusted at 100 W for 2 min, fixed in formaldehyde solution (4% *w*/*v*) and then filtered using white polycarbonate membranes (0.2 μm, Ø47 mm). Then, 4,6-diamino-2-phenylin doldihydrochloride dilactate (DAPI) (Sigma-Aldrich, San Luis, MO, USA) staining (1 mg mL^−1^) was added [39]. Samples were taken in triplicates and biofilms were evaluated on the basis of bacterial cells counted, using a Motic BS-310 epifluorescence microscope equipped with a MotiCam Pro 285A digital camera (Motic, Richmond, Canada).

### 2.10. Oxidation of As(III) to As(V) by the Selected Strain under Planktonic and Immobilized Conditions

The oxidation of As by the selected strain under planktonic and immobilized conditions was studied using 500 mL glass bottles, as batch bioreactors, containing 250 mL CDM plus 0.5 mM sodium arsenite (NaH_2_AsO_3_). The free cells (planktonic) bioreactor was inoculated with 10^6^ CFU mL^−1^ of the selected strain. The immobilized cells bioreactor was inoculated with 10 g of a biofilm (10^8^ cells/g) of the selected strain obtained after 72 h previous incubation on active carbon at 25 °C [40]. Both batch bioreactors were incubated under aerobic conditions at 25 °C during 48 h. The experiments were performed in triplicate and oxidation of As(III) to As(V) was determined from culture supernatants aseptically removed from the reactors. One mL supernatant samples were filtered through a sterile 0.22 μm pore size filter (Corning Inc., New York, NY, USA) and As-species were detected by means of high-performance liquid chromatography coupled to arsine gaseous formation performing the detection by atomic absorption (HPLC/HG/QAAS) [19]. The quantitative determination of both arsenic species (As(III) and As(V)) was performed using an atomic absorption spectrometer (AAnalyst 100, PerkinElmer, Norwalk, CT, USA), equipped with a commercial hydride generation system (FIAS-100, PerkinElmer, Norwalk, CT, USA) and a quartz cell (15 cm length, 1 cm i.d.) associated to an acetylene flame served as atomiser. An externally controlled by An EDL power supply system (EDL System 2, PerkinElmer, Norwalk, CT, USA) allowed to control an electrode-less arsenic lamp used as emission source [19].

### 2.11. Statistical Analysis

Bacterial biotransformation and growth rates (K) of the models that best conformed to the kinetics of bacterial growth were analysed through Student’s t tests using the MINITAB software (version 15, USA). *P* values < 0.05 were considered as statistically significant.

## 3. Results

### 3.1. Sequencing Data and Diversity Estimates

The sequencing analysis of the universal V1-V2 region of the 16S rRNA for Bacteria Domain produced a total of 295,530 sequences. After quality check within the SILVA database and removing chimeras, 294,104 (99.4%) high quality sequences remained (Table 1). The Shannon diversity index and non-parametric Chao1 and ACE richness estimators are reported in Table 1, showing a normal diversity.

### 3.2. Composition of the Bacterial Community

The OTUs retrieved from the Camarones river superficial sediment sample showed the presence of a total of 17 different bacterial phyla. The dominant taxonomic groups (abundance ≥ 1%) were affiliated to *Proteobacteria* and *Firmicutes* (76.6% and 21.9%, respectively) (Figure 1). Within *Proteobacteria*, the most abundant class was *Gammaproteobacteria* (73.43%) and another four classes were also present, including two classes representing minor components (0.44% and 0.2%) (Table 2). Among the phyla whose abundance was ≤1%, OTUs affiliated to fifteen different phyla, with abundances varying from 0.46% (*Acidobacteria*) to 0.02%, were identified (Table 2).

### 3.3. Isolation and Characterization of Bacterial Arsenic-Resistant

After culturing in the presence of As, a total of 11 different As-resistant strains were isolated from the sediment. All bacterial strains isolated were able to grow on agar containing either 7 mM As(III) or 20 mM As(V) at 25 °C, concentrations allowing to considered these strains as As-resistant [35].

One of the bacterial strains, labelled as VC-19, was able to grow at As concentrations exceeding 20 mM As(III) and 100 Mm As(V). Sequencing of PCR product (1200 bp) demonstrated the presence of the arsenite oxidase *aio* gene in this strain. The biochemical profile of this strain and the closest GenBank match to its 16S rDNA sequence revealed that VC-19 strain (GenBank Accession Number MG545624) possessed a high degree of similarity (99.8%) with *Pseudomonas migulae*. Moreover, the tree reconstruction affiliated VC-19 strain to *Pseudomonas* genus, ratifying that the analysed sequence of this strain can be ascribed to *P. migulae* (Figure 2).

### 3.4. Growth Kinetics

In order to determine the effect of As on the growth of *P. migulae* VC-19 strain, its growth kinetics was studied in the presence of 0.5 mM As(III) or As(V) or its absence by means of the Gompertz model. The model demonstrated that after of 48 h of incubation this bacterial strain cultured in the presence of 0.5 mM As(III) showed a better growth than the control, with a growth rate (µ_m_) of 0,88 and 0,79 h^−1^, respectively (Figure 3). Student’s t test (with 95% confidence) demonstrated statistically significant differences between both conditions (*p* < 0.05). When cultured in the presence of 0.5 mM As (V), *P. migulae* VC-19 strain showed a growth rate (µ_m_) of 0.78 h^−1^, significantly lower than in the presence of As (III) Figure 3).

### 3.5. Biofilm Formation in a Batch Experiment by P. migulae VC-19

The ability of 10^5^ CFU mL^−1^ of *P. migulae* VC-19 strain growing in the exponential phase to adhere to activated carbon in the presence or absence of As(III) was monitored by fluorescence microscopy and SEM. The number of bacterial cells per gram of carbon was evaluated up to 72 h of culture by fluorescence microscopy. Both, with or without As(III), the bacterial strain formed a stable biofilm containing 10^9^ and 10^8^ cells g^−1^ of activated carbon, respectively (Figure 4). *P. migulae* VC-19 strain, cultured in the presence of As(III) showed similar growth rate (µ_m_) than the control, 0,60 [CFU h^−1^] under both conditions (*p* > 0.05). In the presence of As(V), the biofilm presented a stable growth, similar to that of the control culture and the culture containing As (III).

Besides monitoring cell counts by fluorescence microscopy, biofilm formation by *P. migulae* VC-19 strain on activated carbon was also monitored by SEM. In the presence or absence of As(III), small cells were randomly distributed on the surface of activated carbon after 24 h of incubation (Figure 5A and Figure 6A). After 48 h of incubation, cultures with or without As(III) showed bacteria anchored to the activated carbon and to each other by means of fibrils, bridging the space between bacteria (Figure 5B and Figure 6B). After 72 h of incubation, both cultures in the presence or absence of As(III), a mature biofilm was observed with attached bacteria associated to extracellular polysaccharides covering the sessile cells and serving as a matrix for further biofilm formation and stabilizing it was observed in both cultures. (Figure 5C and Figure 6C).

### 3.6. Arsenic Oxidation in Batch Experiments

Arsenite oxidation by planktonic or immobilized *P. migulae* cells was evaluated by HPLC/HG/ASS. Immobilized *P. migulae* cells were able to oxidize 0.5 mM As(III), under aerobic conditions, leading to 100% As(III) conversion into As(V) after 36 h of incubation, a rate of 12.86 μg mL^−1^ h^−1^ (Figure 7).

Planktonic *P. migulae* cells, cultured under similar conditions, were able to oxidize 100% As(III) to As(V) after 48 h, a rate of 10.07 μg mL^−1^ h^−1^ (Figure 8). A decrease of total-As present in the supernatant, concomitant with the decrease of As (V), was also observed in the batch experiments implemented with immobilized cells (Figure 7).

## 4. Discussion

Certain aquatic environments are characterized by high concentrations of different minerals. Because of its geological characteristics, the Andean plateau of the Atacama Desert (Chile), including the Camarones river, naturally possesses high concentrations of As (1000 ug/L) [19,20]. In particular, the presence of As in the Camarones river turns it a potential source to isolate bacteria possessing strategies to survive in an As rich environments [41]. These strategies may include As bioaccumulation, biomineralization, biosorption, and biotransformation [42]. There are reports describing those bacterial communities in As-rich groundwater and surface waters are dominated by *Firmicutes, Alpha-, Beta-, Gamma-* and *Epsilon-proteobacteria* [43,44,45,46,47]. In our study, a massive parallel sequencing analysis (Illumina) revealed a significant shift in the bacterial communities of the As contaminated Camarones river sediment studied, where *Proteobacteria* and *Firmicutes* were dominating groups (≥1% of total classified sequences). Similar results were reported by Leon et al. [48], showing that the Camarones river, also part of the Atacama Desert, possessed a higher relative abundance of *Proteobacteria* (40.3%), *Firmicutes* (24.82%) and *Acidobacteria* (12.02%). The bacterial diversity reported by that report was also greater than the one found in our work. The greater diversity reported for the Camarones river may be the consequence of the “altiplanico winter” which causes very intense precipitations in this zone, resulting in a severe downstream flow of water of high turbidity which carries suspended solids [49].

Many reports describe a variety arsenic-resistant and transforming bacterial species from aquatic environments and soil, being *Proteobacteria* species predominant [50,51]. Some culturable bacterial species representative of these *Phylum,* such as those belonging to the *Pseudomonas* genus, have been previously isolated from several environments, including drinking water, and have demonstrated the ability to grow in higher As concentrations. In our study, *P. migulae* VC-19 was able in grow in the presence of 20 mM As (III) and 100 mM As (V). Dey et al. [52] reported the isolation of twelve colonies from ground-water resistant to 100–5000 ppm arsenate and 100–1000 ppm arsenite. In addition, other bacterial genera, such as *Bacillus, klebsiella, Dienococcus, Acidthiobacillus* and *Desulfitobacterium* have been reported as arsenic resistant [53].

Since the growth curve of *P. migulae* VC-19 strain cultured in the presence of As (III), when modelled by the Gompertz function, showed to grow better than the control. Numerous heterotrophic As(III)-oxidizing bacteria, such as *Hydrogenophaga* sp. NT-14 and *A. tumefaciens* GW4, have been reported to capable to obtain energy from the oxidation of arsenite, using the metalloid as the electron source and carbonate as unique carbon source [21,40,54,55,56,57,58,59,60,61,62,63]. However, As (III) oxidation by heterotrophic bacteria is generally considered to be a detoxification mechanism rather than one that can support growth [64]. Zhang et al. [65] reported a novel chemoautotrophic As(III)-oxidizing bacterium, isolated from an As-contaminated paddy soil, able to derive energy from the oxidation of As(III) to As(V), under both aerobic and anaerobic conditions using O_2_ or NO_3_^−^ as electron acceptor.

When we studied the As (III) genetic resistance of *P. migulae* VC-19, by means of PCR and subsequent sequencing, the presence of the As oxidizing *aio* gene was detected. Structurally, the *aio* codified arsenite oxidase is composed of the minor AioB subunit and the main AioA subunit (90–100 kDa), considered a molybdoprotein. The minor subunit is considered to be the main responsible for the interaction with the electron transport proteins (cytochrome). Since the genes encoding these cytochromes are expressed together with the *aioAB* genes, both genes are generally in the same operon [25]. In addition, the oxidation of highly toxic arsenite to less toxic arsenate encoded by arsenite oxidase *aio* gene is a key step of the As-detoxification mechanism by microorganisms [66,67].

*P. migulae* VC-19 strain was able to produce a stable biofilm on active carbon in the presence or absence of As(III). After 72 h of incubation, SEM analyses showed both in presence or absence of As(III), micro-fibrillar structures and bacterial cells embedded by a matrix probably constituted by extracellular polysaccharides, which has been previously described for other microorganisms [68], and which are characteristic of a mature bacterial biofilm, in which extracellular polysaccharides are secreted by attached bacteria and serve as a matrix for the attachment of other cells [69]. In addition, Decho [70] describe that biofilm-mediated bioremediation presents advantage to bioremediation with planktonic microorganisms because cells in a biofilm have a better adaptation, survival and high microbial biomass. Zeng et al. [71] describe that the biofilm formation significantly promoted the bacterial resistance to arsenic.

Our study demonstrated that the immobilization of the *P. migulae* VC-19 strain on activated carbon allowed the development and maintenance of a biofilm, which showed increased oxidation rates than that of planktonic cells. Simeonova et al. [72], studying *Herminiimonas arsenoxidans* immobilized on calcium alginate, demonstrated that immobilized cells oxidized arsenite faster than planktonic cells. Valenzuela et al. [40] reported similar results studying the oxidation of arsenite to arsenate by *Pseudomonas arsenicoxydans* cells immobilized on zeolite. Ito et al. [73] reported in continuous experiments, conducted using a bioreactor with immobilised arsenite oxidising bacteria, an As(III) oxidation efficiency of 92%, with an As(III) oxidation rate of 1 × 10^−9^ μg/cell/min.

Wan et al. [74] investigating the combined processes of biological As (III) oxidation and removal of metalloid by zero-valent iron, reported an incomplete As removal, associated to kinetic limitation of As removal or by a continuous formation of iron corrosion products, enable to trap As. In our study, the total arsenic decreases over time, due to a possible adsorption of the metalloid on the activated carbon used as support for the *P. migulae* VC-19 biofilm. Adsorption onto activated carbon has been described as one of the most effective methods for heavy metal removal [75,76,77]. In general, microbial oxidation of As (III) in bioreactors with immobilized biomass, followed by adsorption is a promising and environment-friendly system for the treatment of As(III)-contaminated waters [78,79,80,81].

## 5. Conclusions

The results of the present study demonstrated that heterotrophic arsenite oxidising bacteria, such as *P. migulae*, are good potential candidates for biological As-detoxification systems. The reason is that As abatement requires, as an initial step, the oxidation of the metalloid for its subsequent precipitation and removal. Therefore, making use of bacteria for the biological oxidation of As may circumvent the requirement of strong chemical oxidants, compounds which are usually even more toxic than arsenic itself.

## Figures and Tables

**Figure 1 microorganisms-09-00483-f001:**
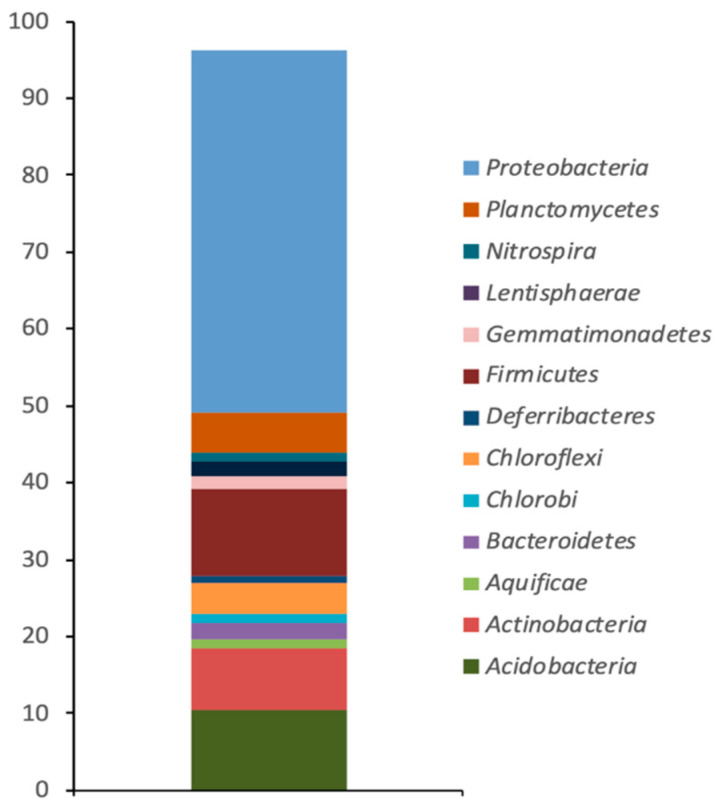
Relative abundance of sequences (percentage) assigned to bacterial phylogenetic groups from the superficial sediment of Camarones river, Atacama Desert.

**Figure 2 microorganisms-09-00483-f002:**
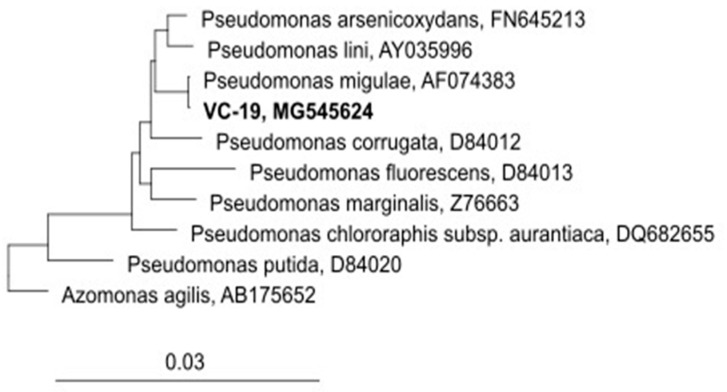
Phylogenetic tree reconstruction based on the rRNA 16s analyses of *Pseudomonas migulae* VC-19 strain. (GenBank N° Access MG545624)

**Figure 3 microorganisms-09-00483-f003:**
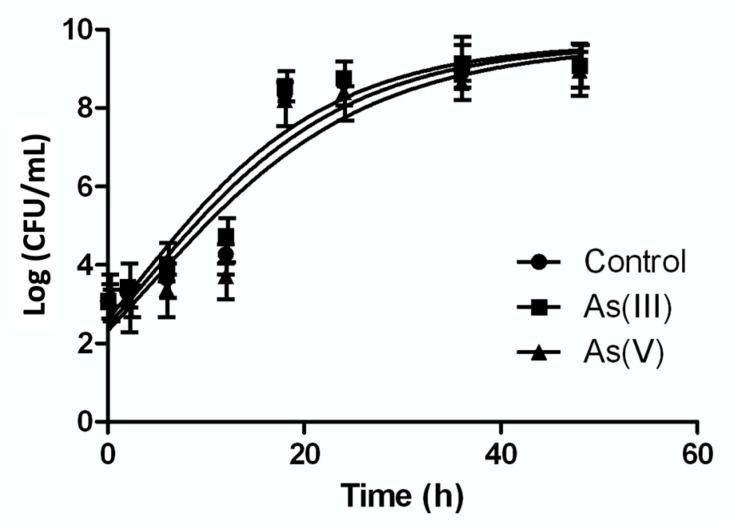
Growth kinetics of *Pseudomonas migulae* VC-19 strain in the presence of 0.5 mM As(III) or 0.5 mM As(V). The same strain cultured in the absence of arsenic was used as control. Kinetics was adjusted with Gompertz growth.

**Figure 4 microorganisms-09-00483-f004:**
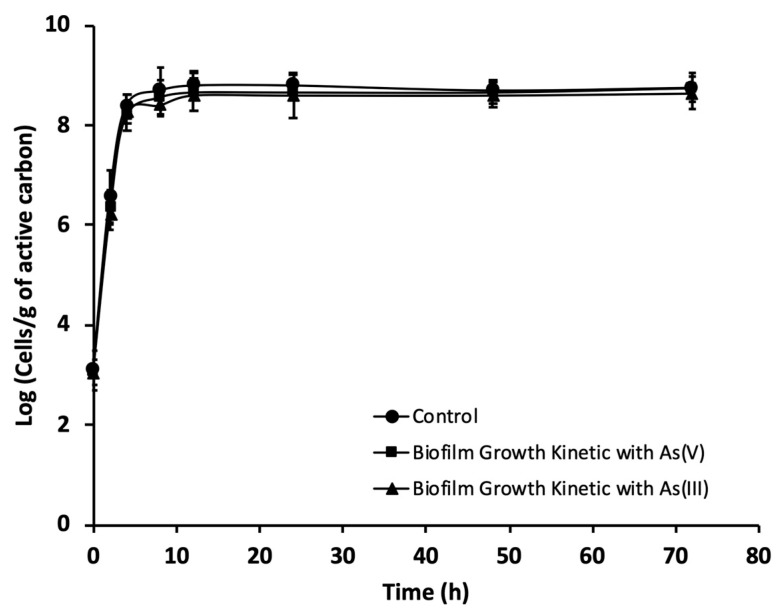
Growth kinetics, during a 72 h period, of *P. migulae* VC-19 on activated carbon plus As(III) or As(V), monitored by means of DAPI staining. The control lacked As-species.

**Figure 5 microorganisms-09-00483-f005:**
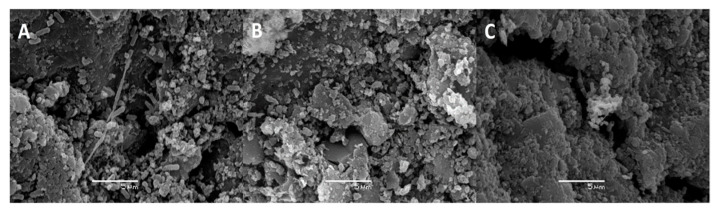
Biofilm formation by *P. migulae* VC-19 in batch experiments containing activated carbon and no Arsenic after 24 h (**A**), 48 h (**B**) and 72 h (**C**) of incubation at 30 °C under aerobic conditions.

**Figure 6 microorganisms-09-00483-f006:**
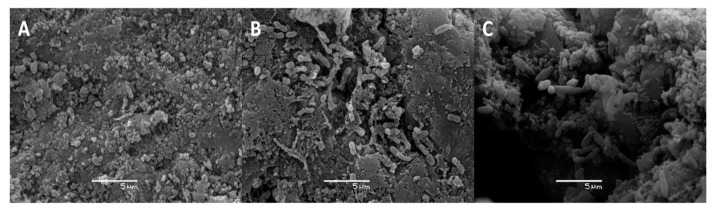
Biofilm formation by *P. migulae* VC-19 strain in a batch experiments containing activated carbon and 0.5 mM As(III) (Na_3_AsO_3_) after 24 h (**A**), 48 h (**B**) and 72 h (**C**) of incubation at 30 °C under aerobic conditions.

**Figure 7 microorganisms-09-00483-f007:**
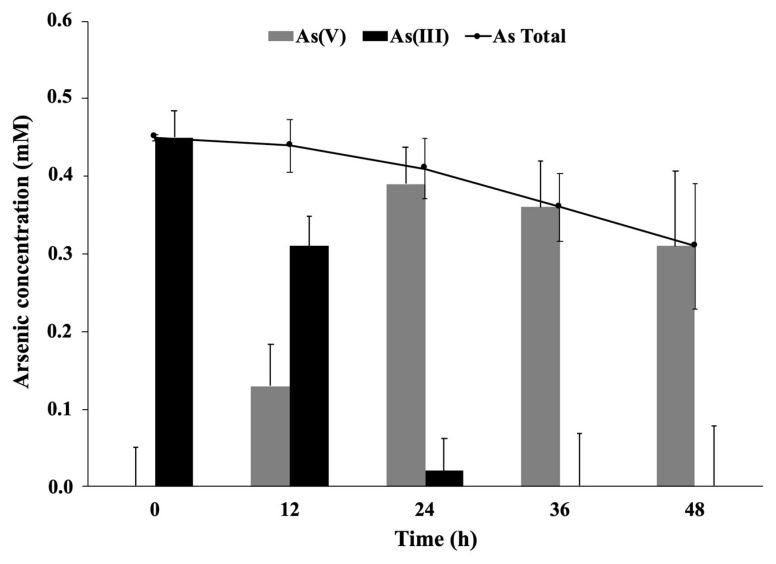
Arsenite oxidation by *P. migulae* VC-19 strain immobilized on activated carbon, cultured at 30 °C during 48 h.

**Figure 8 microorganisms-09-00483-f008:**
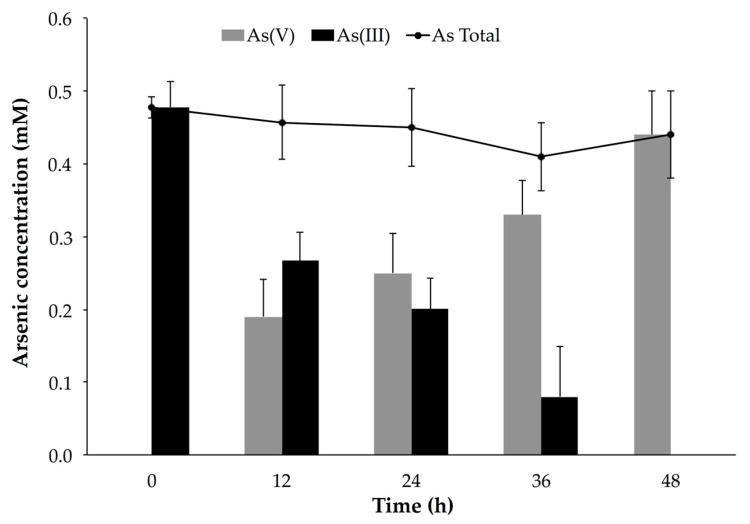
Arsenite oxidation by *P. migulae* VC-19 strain in planktonic conditions at 30 °C, during 48 h.

**Table 1 microorganisms-09-00483-t001:** Sequencing information, diversity index (H’) and estimator of richness (Chao1 and ACE) obtained after Illumina sequencing.

Number of Reads	295,530
Number of High-Quality Reads	294,104
Unique Reads	20,236
% Unique Reads	6.88%
Shannon (H’)	2.365
OTUs at 97% (genetic sim)	4520
Chao1 (TC%)	77,582.439
ACE (TC%)	86,400.014
Number of Bacteria Sequences	18,080

**Table 2 microorganisms-09-00483-t002:** Dominant phylogenetic groups present in the sediment of Camarones river, Atacama Desert.

	Phylogenetic Group	Relative Abundance (%)
**Dominant Phylogenetic Groups (≥1%)**	*Proteobacteria*	76.57
	*Zetaproteobacteria*	0.02
	*Alphaproteobacteria*	1.33
	*Betaproteobacteria*	1.26
	*Deltaproteobacteria*	0.44
	*Gammaproteobacteria*	73.43
	***Firmicutes***	**21.92**
**Dominant Phylogenetic Groups (≤1%)**	*Acidobacteria*	0.46
	*Actinobacteria*	0.35
	*Bacteroidetes*	0.11
	*Chloroflexi*	0.11
	*Tenericutes*	0.09
	*Nitrospirae*	0.09
	*Thermodesulfobacteria*	0.07
	*Gemmatimonadetes*	0.04
	*Planctomycetes*	0.04
	*Aquificae*	0.02
	*Caldiserica*	0.02
	*Acetothermia*	0.02
	*BRC1*	0.02
	*Cyanobacteria/Chloroplast*	0.02
	*Omnitrophica*	0.02

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
