# Peer review of "Characterization of Arsenite-Oxidizing Bacteria Isolated from Arsenic-Rich Sediments, Atacama Desert, Chile"

_microorganisms, 2021, doi:10.3390/microorganisms9030483_

Round 1

Reviewer 1 Report

Authors report an interesting study performed with Camarones river sediments. They define the composition of its bacterial community and isolate some arsenic resistant strains. Specifically, they select VC-19 strain for being the most resistant one and they carry out some experiments in order to characterize its growth kinetics and its ability of forming biofilms in batch experiments. They analyze its ability of oxiding arsenite species both in planktonic assays and after immobilizing them on activated carbon with the aim of developing in the future an environment-friendly system for the treatment of As(III)-contaminated waters.

Manuscript has been properly written and results are shown in a clear way. However, some minor mistakes were detected along the text:

Line 82: Hence, bacteria capable TO oxidize (preposition is missing)

Lines 112 (One gram) and 160 (one g): You should homogenize the form in which you referred to one gram along the text.

Line 222: Isolation (capital letter is missing)

After reading carefully the manuscript, I have some important questions/suggestions for authors:

  • Were the sediments samples maintained using RNA later or other similar product to avoid its degradation? Or were they just maintained at 4ºC as it is mentioned in the text?

  • You could mention in a briefly way the gene aio in your introduction when you talk about bacteria that are able to oxidize arsenic as you will try to detect it then in your experiments.

  • Why do you fix your arsenic concentrations in 7 mM for As (III) and 20 mM for As (V) respectively for defining As-resistant bacteria? Do these values also correspond to the real concentrations that we can find in your Camarones river samples? You may mention the As concentrations that you can find in your samples in the text (maybe in line 301). Moreover, you then select a concentration of 0.5 mM for both arsenic species to continue with your experiments… why do you select this concentration values?

  • It could be interesting to perform a growth kinetic study with different concentrations of As (III) and As (V) for VC-19 strain in order to define its real resistance to arsenic as you previously mention in the text that it is able to resist to concentrations exceeding 20 mM As (III) and 100 mM As (V), that are really higher than the ones selected for the growth kinetic study (0,5 mM).

  • Lines 351-353: I do not really understand why you say that your VC-19 immobilized cells have higher oxidation rates than planktonic ones as no arsenite is detected after 24h in the case of planktonic assays but you still detect arsenic after 36h in the case of immobilized cells (Figures 7 and 8). Maybe the bottom of the figures are not correct or they have been exchanged?

  • Some batch experiments could be performed with real water samples after filtering them with the aim of knowing if your VC-19 strain is able to resist or not to real arsenic values found in them and the percentage of As oxidation that you can detect in this case.

Author Response

Thank you very much for the reviews. We have checked all the remarks made. The response and modifications according their remarks are listed below in blue (Word file). In addition we have left the modifications with changes-control, according to suggested.

Reviewer 2 Report

I found this manuscript great. It is a work of classical microbiology with a lot of novel tools and with the aim to find or identified a strain with a specific characteristic. Great work 

Author Response

The reviewer did not ask questions.
We appreciate the review and your comments.